# Recent Occurrence of PAHs and n-Alkanes in PM2.5 in Seoul, Korea and Characteristics of Their Sources and Toxicity

**DOI:** 10.3390/ijerph17041397

**Published:** 2020-02-21

**Authors:** Minkyung Kang, Kiae Kim, Narae Choi, Yong Pyo Kim, Ji Yi Lee

**Affiliations:** 1Environmental Science and Engineering, Ewha Womans University, Seoul 03760, Korea; kcotek@gmail.com (M.K.); rldo428@gmail.com (K.K.); naraechoi1990@gmail.com (N.C.); 2Chemical Engineering and Materials Science, Ewha Womans University, Seoul 03760, Korea; yong@ewha.ac.kr

**Keywords:** PAHs, n-alkanes, diagnostic ratio, source characteristics, toxicity

## Abstract

Polycyclic Aromatic Hydrocarbons (PAHs) and n-alkanes in particulate matter with an aerodynamic diameter of 2.5 micrometers or less (PM2.5) were quantified at Seoul, Korea in 2018. The seasonal differences in the total concentration of PAHs and n-Alkanes were clear, where winter showed a higher concentration than that of summer. Compared to the PAHs measurements in 2002 at Seoul, the sum of PAHs concentrations in 2018 were reduced from 26.6 to 5.6 ng m^−3^. Major sources of the observed PAHs and n-alkanes were deduced from various indicators such as diagnostic ratios for PAHs and Cmax, CPI, and WNA (%) indices for n-alkanes. It was found that in winter coal and biomass combustions, and vehicular exhaust were major sources, while, in summer vehicular exhaust was major source. In addition, in winter, major emission sources were located outside of Seoul. The health effect from the recent level of PAHs was estimated and compared to the previous studies observed in Seoul, and it was found that, recently, the toxicity of PAHs in PM_2.5_ was significantly decreased, except for in the winter.

## 1. Introduction

Polycyclic Aromatic Hydrocarbons (PAHs) are carbon-structured organic compounds that consist of two or more benzene rings. They are not emitted as a single species, but as a mixture, and their sources are incomplete combustion of fossil fuels such as vehicle emissions, coal combustion, and biomass burning [1,2]. PAHs are significant indicators relating to the adverse health effects of particulate matters (PMs) due to their carcinogenic properties. The United States Environmental Protection Agency (EPA) has listed 16 PAHs as priority pollutants that adversely affect human health; therefore, the control of PAH sources is important to reduce the adverse health effects of PM. The European Union (EU) has also established a health-based standard for PAHs which targets PAHs concentration in ambient air to 1 ng m^−3^, expressed as concentration of Benzo[a]pyrene (BaP) [3]. China has a standard for Toxic Equivalent Quotient (TEQ) of BaP, whereby it must be controlled to 2.5 ng m^−3^ daily and an annual average to 1 ng m^−3^ [4]. However, an established standard for PAHs in South Korea has not been developed.

n-Alkanes are also carbon-structured organic compounds like PAHs, but they are linear carbohydrates which have both anthropogenic and biogenic sources. Fossil fuel combustion and biomass burning are the main contributors to anthropogenic n-alkane sources, and plant wax emission, pollen, and microorganisms largely contribute to biogenic n-alkane sources [5,6,7,8,9].

There have been several studies reporting particle-bound PAHs in the ambient air in South Korea [10,11,12,13,14]. These studies reported the level of PAHs in PM and the possibility of long-range transport of PAHs using back trajectory [15,16]. In addition, some of the previous studies applied receptor models such as the Chemical Mass Balance (CMB) and Positive Matrix Factorization (PMF) models for the source apportionment [16,17]. Coal combustion and biomass burning were identified as a major source of PAHs by the CMB model in the winter of 2002 to 2003 in Seoul, with vehicular emission being the largest PAH contributor for the entire sampling period [16]. The PMF model was applied in Seoul, South Korea, between 2002 and 2013, which found that the contribution of coal combustion decreased, while that of vehicular sources increased during the study period; however, the contribution from biomass burning remained consistent [17].

Numerous studies for PAH characterization in PM have been conducted in various sites worldwide with health risk assessments. Especially, since the emission of fossil fuel combustion in northeast Asian countries such as China and Mongolia has been increasing, recent studies have focused on the health risk assessment of PAHs in PM regarding carcinogenic properties. The identified sources are coal combustion sources and vehicular emissions, and such emission sources are dominant in the colder season [18,19,20,21]. In the city center of Ulaanbaatar City, Mongolia, the lifetime inhalation cancer risk for children and adults in the heating season reached 1.00 × 10^−5^, exceeding EPA guidance value by 10 times [19,22]. BaP-TEQ, estimated at several urban areas in China, ranged from 3.31–4.95 ng m^−3^, which is higher than China’s annual average standard 1 ng m^−3^ [3,19]. However, health risk assessment for PAHs in PM has not been estimated in Seoul, Korea based on recent concentrations of PAHs.

The general approach to classify the sources of PAHs is using a diagnostics ratio. Diagnostics ratio is defined with a pair of PAHs that have similar properties such as molecular weight or formula, but have different physical characteristics such as decay rates. Emission sources can also be identified simply through observing the ratio of individual PAHs. At low temperature, Low Molecular Weight (LMW) PAHs are formed, while at higher temperatures, High Molecular Weight (HMW) PAHs with five or more benzene rings form easily [23]. In addition, combustion of solid fuels such as coal and biomass is a source for LMW and Middle Molecular Weight (MMW) PAHs and liquid fuel combustion such as vehicular emissions is a source for MMW and HMW PAHs [23,24,25,26].

To estimate source characteristics of n-alkanes, three indexes are mainly applied to quantify which emission source is dominant. Carbon Preference Index (CPI) is calculated by the ratio of the sum of odd numbered n-alkanes to the sum of even numbered n-alkanes. When n-alkanes are emitted solely from anthropogenic sources, the CPI will converge to 1, but the effect of biogenic sources will bring the value closer to 3 or more [27,28,29]. Second is C_max_, which is the carbon number of the most abundant species among the extracted n-alkanes. The last index is Wax n-alkanes percentage (WNA, %), which represents the total percentage of plant wax contribution to the n-alkane concentration [30].

Principle Component Analysis (PCA) is an effective statistical tool to identify independent factor of air pollutants by grouping chemical species which share similarities of variances of the species to give physicochemical significance to these groups. PCA has been widely used to the sources identification of air pollutants in PM_2.5_ [31,32].

The objective of this study is to (1) evaluate the recent PAH concentrations in Seoul, South Korea with seasonal characteristics of PAHs, and (2) identify the major sources of PAHs in this area based on the PCA with the various diagnostic ratios of PAHs and index of n-Alkanes. Finally, (3) estimate the toxicity of PAHs in PM_2.5_ in Seoul, Korea in order to provide the basis for the establishment of the ambient air quality standard for PAHs in atmospheric PM in Korea.

## 2. Materials and Methods

### 2.1. Sampling

Samples were collected from the rooftop of Korea Environmental and Technology Institute (37°61′ N, 126°93′ E) located in Seoul, South Korea. The sampling site was adjacent to a road in the south, a residential area in the north, and Mt. Bukhansan in the northeast. Figure 1 is a map of the sampling site.

The sampling period was from January 15th 2018 to December 8th 2018, and at least 23 samples were collected for each season. Detailed information on the number of samples and meteorological parames during the sampling period are given in Table 1. Average radiation, temperature and relative humidity (RH) was at maximum in summer. Radiation for winter and fall was similar, however, average temperature for winter (−4.7 ± 6.1°C) was below the freezing point. RH increases from winter to summer. Samples were collected on pre-baked (at 550 °C for 12 h) quartz filters (QFFs, Pall, Tissuequartz-2500QAT-UP, 8 × 10in) with PM_2.5_ high volume samplers (SIBATA, HV-RW and Tisch 6070V-2.5). Samplers were operated at a flow rate of 1000 L min^−1^ for 23 h for each sample.

### 2.2. Analytical Procedure

A quarter of QFFs were used for each sample extraction. For PAH and n-alkane quantification, internal standardization was applied. The surrogate standards for PAHs were phenanthrene-d10, fluoranthene d-10, chrysene-d12, perylene-d12, and benzo[ghi]perylene-d12, and for n-alkanes, three surrogate standards (C24-d50, C30-d62, and C36-d74) were used. The mixture of the surrogate standards was added to the filters and placed in a 125 mL amber vial. Particulate PAHs and n-alkanes were then extracted by ultrasonication with DCAM/MeOH (3:1, v/v) solvent twice for 30 min. Then, solvents were evaporated with an evaporator (TurboVaP Ⅱ, Caliper) at 40 °C until the total volume of the extracts reached 10 mL. The extracts were filtrated with 0.45 μm pore size filters (Acrodisc 25mm Syringe Filter, Pall), then concentrated to 500 μL at 40 ℃ with gentle N_2_ gas stream using a needle concentrator (#TS-18821, Reacti-therm). In this study, 14 PAH compounds and 17 n-alkanes were identified using an Agilent 7890B gas chromatograph (GC) with DB-5MS ultra inert column (30 m × 0.25 mm, 0.25 µm thickness, Agilent^®^ J&W™) and quantified with a 5977A mass spectrometer (MS) in synchronous selected ion monitoring (SIM)/scan mode. Helium of 99.999% purity was used as a carrier gas at a flow rate of 1mL min-1 in a GC-MS. A 1μL sample was injected in splitless mode at 240 °C. Mass spectrometry was operated in electron impact (EI) mode at 70 eV at a source temperature of 230 °C.

The quantified compounds and their abbreviations are stated in Table 2. The extraction recovery for PAHs was in the acceptable range from 83% to 115%, except Cor. The recovery of n-Alkanes was in the acceptable range from 75% to 129%, except C20. Due to the lowest and highest volatiles of Cor in PAHs and C20 in n-Alkanes, the recovery of Cor and C20 was not stable (Table 2).

## 3. Results

### 3.1. The Concentrations of PAHs and n-Alkanes in PM2.5

The annual and seasonal concentrations of total PAHs and n-alkanes are shown in Table 3. Total PAH and n-alkane concentrations for the whole sampling period were 5.6 ± 7.9 ng m^−3^ and 17.1 ± 13.1 ng m^−3^, respectively. n-Alkane concentration was several folds higher than that of PAHs during the whole period. The winter season recorded the highest average concentration for both compounds (PAHs 16.1 ± 10.01 ng m^−3^; n-alkanes 28.4 ± 16.4 ng m^−3^) and the lowest concentration in the summer season for both compounds (PAHs 0.8 ± 0.5 ng m^−3^; n-alkanes 6.4 ± 2.5 ng m^−3^). The average concentration of PAHs for fall (3.8 ± 2.8 ng m^−3^) was higher than for spring (1.8 ± 12 ng m^−3^), while n-alkanes for spring (16.4 ± 12.5 ng m^−3^) and fall (16.3 ± 7.1 ng m^−3^) did not significantly differ from each other.

Compared with the previous results measured between 2002 and 2003 in Seoul [16], recent annual averaged PAH concentrations in Seoul dropped from 26.6 ng m^−3^ in 2002 and 2003 to 5.6 ng m^−3^ in 2018 to be 1/5 in 2018 compared to 2002 and 2003. There are two possible reasons for the decreased PAH concentrations in Seoul. One is the increase of combustion efficiency of fossil fuel from emission sources of PAHs in Korea. Evidence to prove the combustion efficiency in Seoul is the decrease of Elemental Carbon (EC) concentration in Seoul, as shown in Table 4, which is an indicator of incomplete combustion. Organic Carbon (OC) and EC measurement have been carried out since 2013 at the same sampling site of this study, operated by the Korean National Institute of Environmental Research (NIER) as one of the intensive measurement networks of the NIER. EC concentration has shown a decreasing tendency since 2013, while, OC concentration has not been changed. Compared to the EC concentration in 2013, the EC concentration in 2018 is about half, suggesting higher combustion efficiency of fossil fuel emission sources. In addition, the OC/EC ratios which is one of indicator of secondary formation of OC in PM2.5 has been doubled from 2013 to 2018. It can be explained by an increase in secondary formation or a decreased of primary emission. The consistency of OC concentration with the increasing OC/EC ratios is primarily due to the effect of the decrease of EC concentration. Therefore, the tendency of EC concentration with OC/EC ratio can be one of evidence for the decrease of organic PM in combustion sources. Also, increase of clean fuel usage instead of coal fuel according to the policy of air quality management in Korea is one of possible reason to reduce PAHs concentration at Seoul, Korea. Another possibility is the decrease of PAH concentrations outside of Seoul, including China. Indeed, the recent PAH concentrations in PM measured in China has decreased significantly in recent years [33] (Table 3 and Table 4).

### 3.2. Seasonal Characteristics of PAHs and n-Alkanes

Table 5 shows the correlation between PAH and n-alkane concentrations for each season. For the whole period, it shows a strong correlation (R > 0.73). Winter samples show the best correlation (R = 0.80), and summer, the weakest correlation (R = 0.30), suggesting that in winter, PAHs and n-alkanes have similar emission sources compared to summer (Table 5).

#### 3.2.1. PAH Seasonal Characteristics

PAHs are classified by rings, as shown in Figure 2. Only the benzene rings are counted as “rings”; the other cyclic carbon frames are considered “0.5 rings”. Figure 2 is the seasonal averaged ratio of 3–7-ringed PAHs. The composition of PAHs by number of benzene rings shows a seasonal difference. The ratio of PAHs with three benzene rings tends to decrease toward summer, while PAHs with four and six benzene rings show the opposite trend. Since the combustion emission source of LMW-PAHs is coal and biomass combustion and HMW-PAHs Vehicle, the summer is more affected by vehicular emissions and less by solid fuel combustion. Also, this tendency is due to the semi-volatile characteristics of 3 and 4 benzene rings of PAHs in the ambient air. (Figure 2).

The five diagnostics ratios (DRs) used in this study to estimate the source characteristics of PAHs are shown in Table 6. Four of them—IcdP/IcdP+BghiP [34,35], Fl/Fl + Py [35,36,37], BaP/BghiP [38,39], BaA/BaA + Chr [37]—were used to separate coal and biomass combustion and vehicular emissions *(*Table 6). As seen in Table 6, separation of coal and biomass combustion is limited using DR of specific PAH compounds due to overlapping DR of biomass burning and coal combustion. So, in this study, the emission sources of PAHs can be only separated into solid fuel combustion, which includes both of coal and biomass combustion with vehicular exhaust emission.

Figure 3a–e shows seasonal box plots of PAH DRs. There is a clear distinction between winter and summer season PAH emission sources, where winter PAH sources are a mixture of of solid fuel (coal + biomass) combustion and vehicular emissions, and summer is more affected with vehicular emissions.

Particle aging was evaluated with the remaining ratio BaP/BaP+BeP [40], shown in Figure 3e. The higher intensity of the photochemical reactions in summer shows well the aged characteristics of the sampled PM. However, particle aging can also be observed in other seasons, including winter, a season that provides the least factors for atmospheric reactions, implying the possibility for PAH transport from outside of Seoul (Figure 3).

#### 3.2.2. n-Alkane Seasonal Characteristics

Shorter chained n-alkanes (C 25) are a known source of fossil fuel combustion. Plant wax emissions contribute to longer chained n-alkanes (C 25) with odd number carbons [28,41,42,43,44,45]. Figure 4a–d shows figures of averaged seasonal concentrations of n-alkanes. In order to compare the distribution of n-alkane concentrations by carbon number, the scale of the y axis for each figure was applied differently. The ratio of carbon chains shorter than C25 was higher in the winter compared to the other seasons, while the odd chained carbons longer than C25 gain importance from winter to summer (Figure 4).

Figure 5 shows the correlation between CPI value and WNA. The correlation value was calculated as R = 0.69, a moderate correlation (0.5 < R < 0.7), meaning CPI and WNA are a good index for source contribution evaluation [46,47,48,49]. In Table 7, the calculated seasonal average CPI values range from 1.43 to 1.97. From winter to fall, CPI is closest to 1 in the winter and spring and increases onwards. WNA increases up to the summer season, then drops in fall. In addition, C27, C29, and C31 n-alkanes peak in all seasons. All of the information suggests that the entire sampling period was affected by plant wax sources, which was more significant in the warmer seasons such as summer, and the anthropogenic combustion sources dominated n-alkane production, especially in the winter compared to the other sampling periods (Figure 5) (Table 7).

### 3.3. PCA Analysis

PCA was performed to group measured species with common variances, i.e., principal components. Normally, the first principal component would represent the highest variance, followed by the second and third, and so on. Species showing higher values for each factor were considered to share an origin. For PCA, the concentration of 14 PAH and 16 n-alkane compounds were applied as variables into the SPSS 18.0 statistical software [50] for 108 total PM_2.5_ samples. Varimax rotation was used to get as many positive loadings as possible to achieve a more meaningful and interpretable solution for air pollutants data suggested from previous studies [31,32]. N-alkanes were accounted into two groups: low (∑C20-C25) and high (∑C26-C36) molecular weight (MW) to characterize fossil fuel combustion and biogenic emission, respectively. For 14 PAHs, each ring-group of PAHs were used as indicator to distinguish the combustion sources of solid fuel (coal + biomass) and liquid fuel (vehicular emission). Combustion of solid fuels such as coal and biomass mainly emit PAHs that have three and four benzene rings, and liquid fuel combustion such as vehicular emissions is a source for PAHs with five or more benzene ring [18,19,20,21]. Sensitivity analysis by number of variable was conducted in PCA and those results were found to be relatively consistent.

Five factors were extracted as the results of eigenvalue larger than 1. Generally eigenvalue larger than 1 is chosen for valuable components, however, the rapid change of slope were appeared at component 4 from scree plot shown in Figure 6. Thus, three components were valuable to identify sources of PAHs and n-Alkanes in PM2.5. Indeed, variance for Factors 4 and 5 were 7% and 6%, respectively, which was insignificant compared to Factor 1 (25%), Factor 2 (21%), and Factor 3 (16%).

Table 8 shows the results of the PCA analysis for total PM_2.5_ samples. Three factors are identified and these account for 62.6% of the variability in the data. Factor 1 was explained as a mixture of vehicle emissions and solid fuel combustion, as the significance of both LMW-PAHs (Phe, Ant, Fl) [21] and HMW-PAHs, such as Ind and BghiP [40,46] with n-alkanes shorter than C25, were grouped together. Contrary to Factor 1, only PAHs with 5~6 rings were highly loaded with shorter chained n-alkanes. Thus, Factor 2 was identified as a vehicular source. Vehicular emissions in factor 1 were separated with Factor 2, which might be related to the difference of the origin of the vehicular emissions for Factors 1 and 2. Factor 3 only highlights n-alkanes over C27, which are markers of biogenic sources [28,41,42,43,44,45] (Table 8).

Figure 7 shows both the loading and score plot of PCA analysis for three factors. Winter and other seasons’(spring, summer, fall) samples were well separated (Figure 7a,b) which indicates different emission sources. The lower right part of the score plot is dominated by winter samples, which is characterized by LMW-PAHs, HMW-PAHs and lower chained n-alkanes in the loading plot (Figure 7b). Therefore a mixture of solid fuel combustion and vehicle emission is a major influence on winter atmosphere. Fall samples mostly overlapped with spring and summer samples in the lower left corner of the score plot, which implies the samples of these seasons has similar emission source. The overlapped area of the three seasons all included HMW-PAHs (DahA, Cor) and longer chained (C>25) in the loading plot. Thus we can expect that spring, summer and fall were mainly influenced by biogenic emissions and vehicular emission

### 3.4. Back Trajectory Result

Figure 8a–d shows the seasonal cluster of three day back trajectory performed with HYSLPLIT, and averaged wind speed and direction calculated with WINDROSE. Since wind information at the PM_2.5_ sampling site was not measured, data from a meteorological station in Seoul (Station ID: 108, 37°34′ N, 126°57′ E) was utilized.

In the HYSLPLIT analysis, five clusters were extracted in the winter season. Along with local wind direction and speed, wind parcels from the Northwest of Korea were dominant. Meanwhile, cluster 1 (24%) and cluster 4 (27%), which was from the East of Korea in summer, took up more than half the fraction of the total cluster. These two clusters might be highly affected by local emission sources. Spring and fall show a pattern mixture of summer and winter. In Spring, 67% of the back trajectory was identified as local emission sources due to air parcels from the East of Korea, and the next abundant cluster, which was from Russia, took up 22% of spring back trajectory. Wind from the west was dominant for local wind direction, but the Northeast frequency increased compared to the previous season. In fall, 20% (Cluster 2) of clusters passed to the east side of the Korean peninsula, local transport, and the rest from the west.

Seasonal differences in PAH emission sources were observed in the diagnostic ratios of the previous section, with winter having relatively dominant coal and biomass combustion sources. The back trajectory and local wind direction results of Factor 1 of the PCA analysis, identified as a mix of coal and biomass combustion and vehicular emissions, support PAHs derived from these sources which likely originated from outside of Seoul, especially the northwest region of China. The summer diagnostics ratio and Factor 2 of the PCA analysis from the previous section indicates a dominance of vehicular emissions, and the back trajectory and local wind direction show the locality of PAH source of summer.

### 3.5. Toxic Equivalency Quotient (TEQ) and Mutagenic Equivalency (MEQ) of PAHs in PM2.5

Since some of the PAH compounds are carcinogen, it is important to estimate the health effect from these compounds to the people in Seoul, Korea due to the high population concentrations in this region. There are two types of methods to estimate health effects due to PAH compounds—one is the toxicity equivalency factor (TEF) [51] and the other is the mutagenic equivalency factor (MEF) [52]. The reference compound is BaP, which is known to be the most harmful PAH compound. The TEF reflects for relative potency of individual PAH compounds to BaP, while the MEF assesses mutagenicity of individual PAH compounds relative to BaP, representing the potential risks for non-cancerous adverse health effects. BaP-TEQ and BaP-MEQ were calculated by multiplying the concentrations of each compound with its TEF and MEF values, respectively, as shown below.

(BaP-TEQ) = {Phen} × 0.001 + {Anthr} × 0.01 + {Flt} × 0.001 + {Pyr} × 0.001 +{B(a)A} × 0.1 + {Chry} × 0.01 + {B(b)F} × 0.1 + {B(k)F} × 0.1 + {B(a)P} × 1 + {D(ah)A} × 5 +{Ind} × 0.1 +{B(ghi)P} × 0.01

(BaP-MEQ) = {B(a)A} × 0.082 + {Chry} × 0.017 + {B(b)F} × 0.25 + {B(k)F} × 0.11 + {B(e)P} × 0.002 + {B(a)P} × 1 + {I(1,2,3-cd)P} × 0.31 + {D(ah)A} × 0.29 + {B(ghi)P} × 0.19

Table 9 shows a comparison of TEQ and MEQ of PAHs in 2002 and 2018 measured in Seoul, Korea. The 2018 observation showed one order lower values of both toxicity and mutagenicity than 2002. The highest average BaP-TEQ was observed in winter in both 2002 (7.15 ± 4.8 ng m^−3^) and 2018 (1.86 ± 2.0 ng m^−3^), 2002 showing about 3 times the toxicity compared to 2018. Summer recorded the lowest BaP-TEQ in both 2002 (1.78 ± 1.0 ng m^−3^) and 2018 (0.08 ± 0.09 ng m^−3^), 2018 being 22 times lower toxicity than 2002.

Since South Korea has not specified a PAH ambient air standard, this study applied the Chinese National Ambient Air Quality Standard (GB 3095-2012) [4] annual standard 1 ng m^−3^. The number of sampling days in 2002 that exceeded this standard was 20 times in winter, 7 in spring, 2 in summer, but none in fall, while in 2018, only 5 days did not meet in winter and for the whole sampling period. The application of WHO’s stricter standard 1 ng m^−3^ [53] added the days that exceeded the standard, as shown in Table 9. However, in 2018, all seasons except winter met the WHO standard.

This result revealed that recent BaP-TEQ in PM_2.5_ in Seoul, Korea satisfied both Chinese NAAQS and WHO’s strict standard except winter. However, still, the toxicity of PM_2.5_ can be increased due to high concentrations of PAHs in PM2.5 during winter. In order to minimize the toxicity (i.e., cancer risk) by PAHs in PM2.5, continuous monitoring of PAHs in PM_2.5_ with establishment of air quality standard is required (Table 9).

## 4. Summary and Conclusions

In this study, we found that recent PAH concentrations in PM_2.5_ in Seoul, Korea were affected by the combination of solid fuel (coal + biomass) combustion and vehicular emissions in the winter season by the diagnostics ratio and PCA result. The n-alkane index supports these results. The back trajectory traces the sources back to the far northwest of the Korean peninsula. While, in summer, vehicle emissions were dominant and cluster analysis indicates that this is mainly due to the local emission effect.

Comparison of 2002 PAH data, recent PAH concentrations in PM_2.5_ (measured in 2018) showed a drastic decrease in both toxicity and mutagenicity. In 2018, BaP-TEQ exceeded the Chinese NAAQS and WHO standard only in winter, a season most affected by the mixture of coal and biomass combustion and vehicular emissions and air parcels traveling from the Northwest region of Korea. Therefore, the control of PM2.5 toxicity in Seoul needs to focus on the regulation of PAH emission sources in the winter season.

## Figures and Tables

**Figure 1 ijerph-17-01397-f001:**
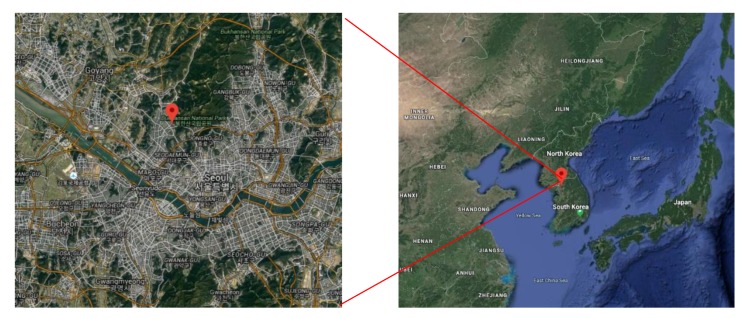
The location of the sampling site is flagged with a red point in the map.

**Figure 2 ijerph-17-01397-f002:**
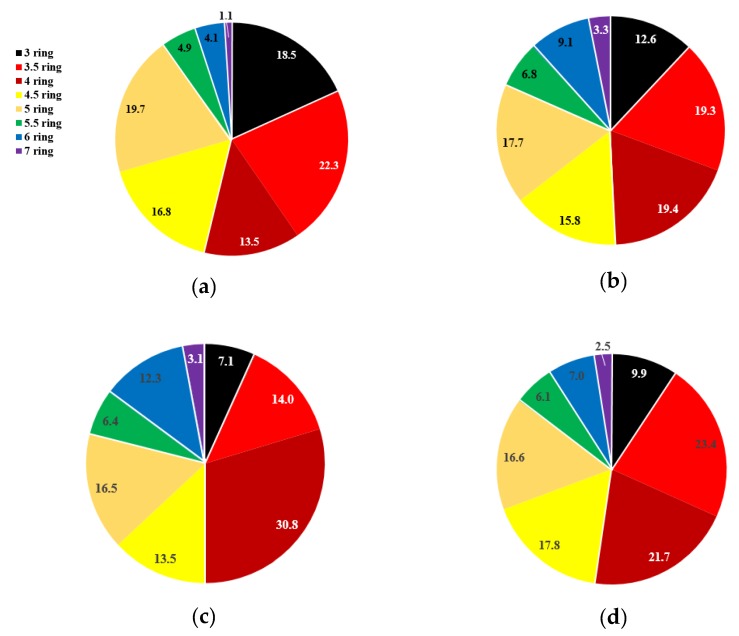
Percent of different ringed PAHs for each season: (**a**) Winter, (**b**) Spring, (**c**) Summer, and (**d**) Fall.

**Figure 3 ijerph-17-01397-f003:**
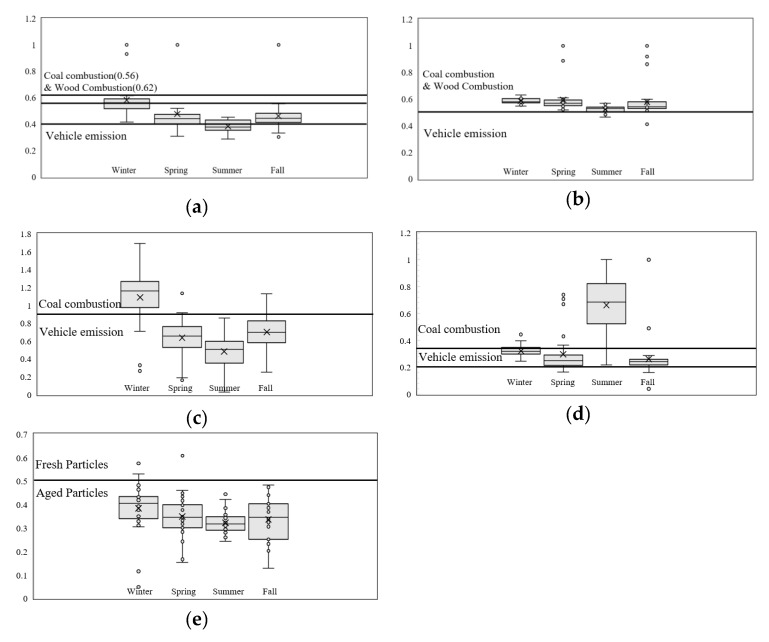
Diagnostics Ratio of PAHs. (**a**) IcdP/IcdP+BghiP; (**b**) FL/Fl+Py; (**c**) BaP/BghiP; (**d**) BaA/BaA+Chr (**e**) BaP/BaP+BeP.

**Figure 4 ijerph-17-01397-f004:**
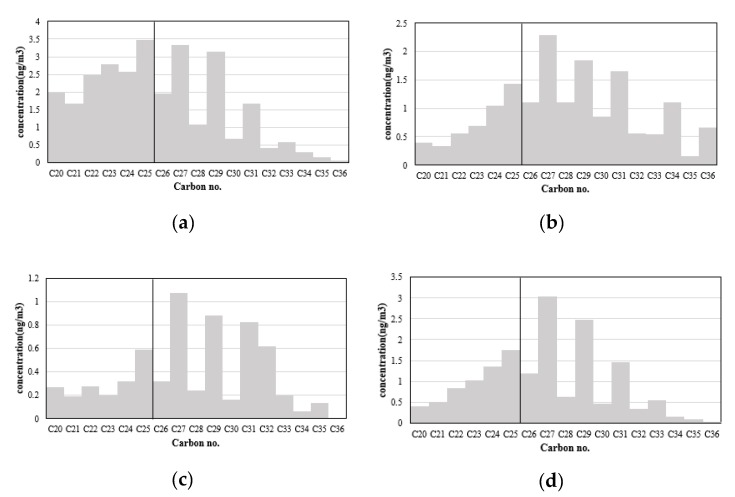
Seasonal averaged n-alkane concentrations. (**a**) Winter; (**b**) Spring; (**c**) Summer; (**d**) Fall. The scale of the y axis for each figure is different.

**Figure 5 ijerph-17-01397-f005:**
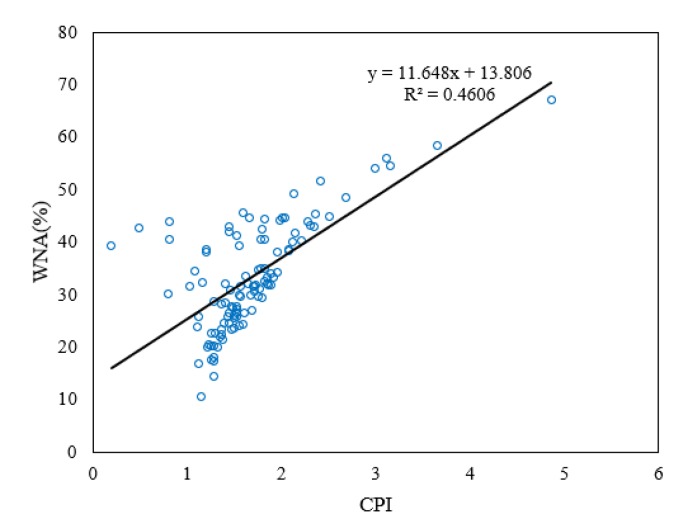
Carbon preference index (CPI) and wax n-alkanes percentage (WNA) correlation Plot.

**Figure 6 ijerph-17-01397-f006:**
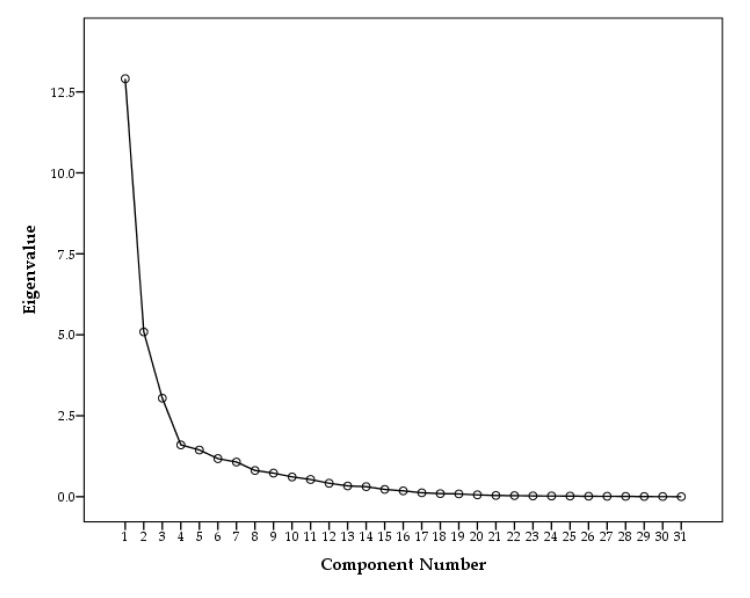
Scree plot of the Principle Component Analysis.

**Figure 7 ijerph-17-01397-f007:**
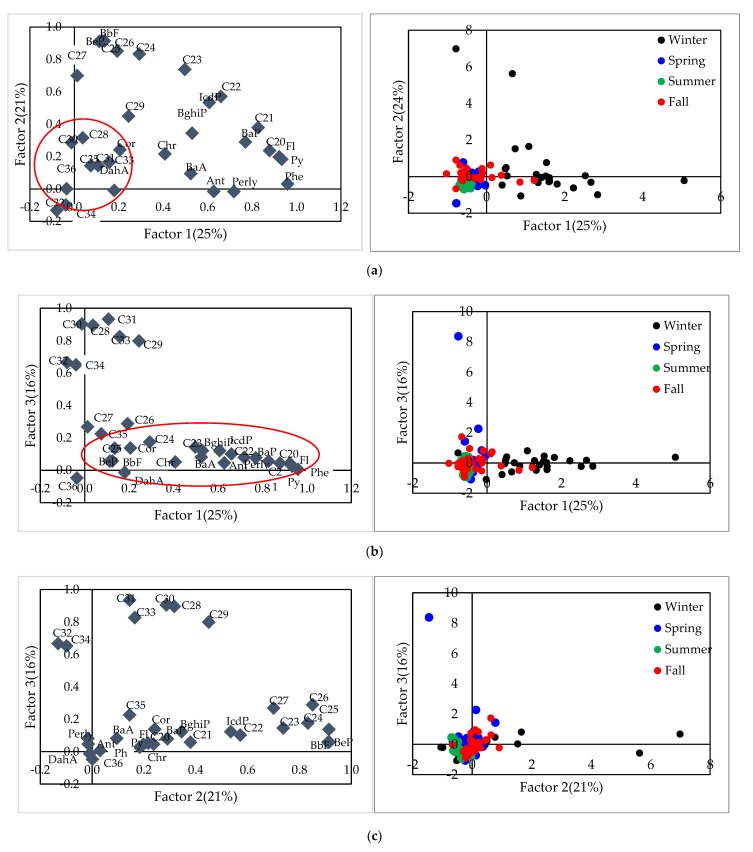
Loading plots and Score plots of (**a**) Factor 1 and Factor 2; (**b**) Factor 1 and Factor 3; (**c**) Factor 2 and Factor 3.

**Figure 8 ijerph-17-01397-f008:**
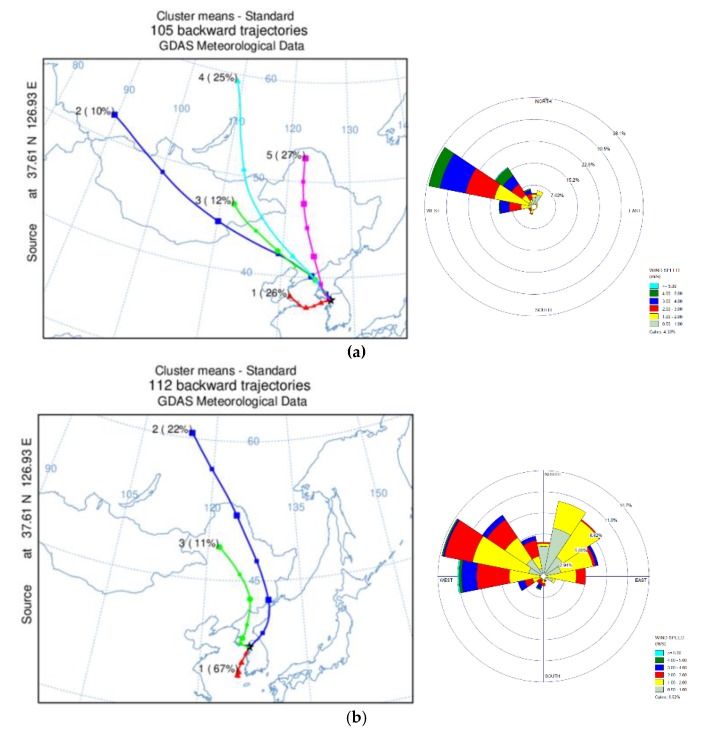
Seasonal back trajectory cluster analysis performed with HYSPLIT and averaged wind direction and speed analysis performed with WINDROSE. (**a**) Winter; (**b**) Spring; (**c**) Summer; (**d**) Fall.

**Table 1 ijerph-17-01397-t001:** Number of samples and meteorological parameters during the sampling period for each season.

	Period	No. of Samples	Radiation (MJ m^−2^)	Temperature (℃)	RH(%)
Winter	2018/01/15 –2018/02/09	25	7.8 ± 3.8	−4.7 ± 6.1	45.8 ± 10.5
Spring	2018/03/11 –2018/04/07	28	13.8 ± 3.8	14.4 ± 4.6	61.4 ± 11.9
Summer	2018/08/20 –2018/09/16	23	14.4 ± 6.7	24.3 ± 2.3	67.2 ± 13.8
Fall	2018/11/05 –2018/12/08	32	7.8 ± 3.4	6.4 ± 5.0	56.7 ± 16.7

**Table 2 ijerph-17-01397-t002:** Quantified Polycyclic Aromatic Hydrocarbons(PAHs), n-Alkanes, and abbreviations and recovery (%).

PAHs Compounds	Abbrev.	Recovery (%)	n-Alkane Compounds	Recovery (%)
Phenanthrene	Phe	89 ± 2.0	C20	59 ± 18
Anthracene	Ant	84 ± 3.0	C21	82 ± 16
Fluoranthene	Fl	88 ± 1.2	C22	105 ± 12
Pyrene	Py	36 ± 1.7	C23	113 ± 7.0
Benzo[a]anthracene	BaA	97 ± 2.0	C24	114 ± 5.0
Chrysene	Chr	92 ± 1.2	C25	117 ± 13
Benzo[b]fluoranthene	BbF	115 ± 14	C26	113 ± 22
Benzo[e]pyrene	BeP	111 ± 13	C27	111 ± 28
Benzo[a]pyrene	BaP	100 ± 9.7	C28	129 ± 29
Perylene	Peryl	91 ± 4.5	C29	118 ± 11
Indeno[1,2,3-c,d]pyrene	IcdP	95 ± 7.6	C30	105 ± 10
Dibenz[a,h]anthracene	DahA	97 ± 7.8	C31	99 ± 17
Benzo[g,h,i]perylene	BghiP	83 ± 1.5	C32	88 ± 21
Coronene	Cor	49 ± 14	C33	79 ± 23
			C34	75 ± 21
			C35	
			C36	

**Table 3 ijerph-17-01397-t003:** Comparison of PAHs and n-Alkane levels with a previous study in 2002 (unit: ng m^−3^).

	This Study		Previous Study ^1^
	PAHs	n-Alkanes	PAHs
Winter	16.1 ± 10.01	28.4 ± 16.4	50.5 ± 32.3
Spring	1.8 ± 1.2	16.4 ± 12.5	21.8 ± 16.1
Summer	0.8 ± 0.5	6.4 ± 2.5	6.8 ± 3.9
Fall	3.8 ± 2.8	16.3 ± 7.1	20.6 ± 24.0
Total	5.6 ± 7.9	17.1 ± 13.1	26.6 ± 28.4

^1^ Data from 2002 study on PAHs in the ambient air of Seoul [16].

**Table 4 ijerph-17-01397-t004:** Average organic carbon (OC) and elemental carbon (EC) concentration with OC/EC ratios in Seoul.

	EC (μg m^−3^)		OC(μg m^−3^)		OC/EC	
Year	Average	SD	Average	SD	Average	SD
2013	1.8	1.1	3.8	2.6	2.2	1.1
2014	1.6	1.0	3.9	2.7	2.9	3.9
2015	1.3	0.7	3.7	2.2	4.0	5.9
2016	1.2	0.9	3.6	2.2	3.6	6.7
2017	1.3	0.8	3.7	2.2	3.8	5.2
2018	0.7	0.4	4.0	2.6	5.6	2.0

Data provided by the Korean National Institute of Environmental Research (NIER).

**Table 5 ijerph-17-01397-t005:** PAH and n-Alkane concentration correlation.

	Total Period	Winter	Spring	Summer	Fall
Correlation	0.73	0.80	0.42	0.30	0.34

**Table 6 ijerph-17-01397-t006:** Diagnostics Ratio (DR) for anlaysis.

	IcdP/(IcdP + BghiP)	Fl/(Fl + Py)	BaP/BghiP	BaA/(BaA + Chr)	BaP/(BaP + BeP)
Coal combustion	0.56 [35]	>0.5 [37,38]	0.9–6.6 [39,40]	>0.35 [38]	
Vehicular Emission	0.18–0.40 [35,36]	0.4–0.5 [36,37,38]	0.3–0.44 [40]	0.2–0.35	
Wood Combustion	0.62 [35]	>0.5 [37]			
Fresh Particle					~0.5 [40]
Aging Particle					<0.5 [40]

**Table 7 ijerph-17-01397-t007:** Seasonal n-Alkane index.

Index	Winter	Spring	Summer	Fall
Cmax	25, 27, 29	27, 29, 31	27, 29, 31	27, 29, 31
CPI	1.43 ± 0.3	1.43 ± 0.6	1.80 ± 0.6	1.97 ± 0.6
WNA (%)	23.8 ± 6.2	28.1 ± 9.8	42.5 ± 6.5	36.8 ± 8.9
Fuel Combustion (%)	76.2	71.9	57.5	63.2

**Table 8 ijerph-17-01397-t008:** Principle Component Analysis(PCA) factor loadings (only the values larger than 0.5 were marked as bold) for the entire sampling period.

	Component		
	1	2	3
Phe	**0.960**	0.031	0.007
Ant	**0.628**	–0.015	0.046
Flt	**0.933**	0.183	0.027
Pyr	**0.923**	0.198	0.044
BaA	**0.524**	0.095	0.082
Chr	0.408	0.216	0.052
BbF	0.124	**0.915**	0.059
BeP	0.113	**0.910**	0.058
BaP	**0.770**	0.290	0.078
Perly	**0.718**	–0.016	0.081
IcdP	**0.607**	**0.535**	0.122
DahA	0.180	–0.009	–0.013
BghiP	**0.530**	0.345	0.122
Cor	0.205	0.242	0.138
C20	**0.878**	0.237	0.045
C21	**0.827**	0.379	0.058
C22	**0.660**	**0.571**	0.102
C23	0.497	**0.738**	0.145
C24	0.293	**0.833**	0.176
C25	0.135	**0.914**	0.137
C26	0.193	**0.851**	0.289
C27	0.013	**0.700**	0.269
C28	0.037	0.317	**0.896**
C29	0.244	0.450	**0.800**
C30	–0.013	0.286	**0.904**
C31	0.107	0.144	**0.934**
C32	–0.080	–0.132	**0.667**
C33	0.157	0.165	**0.826**
C34	–0.039	–0.099	**0.652**
C35	0.074	0.144	0.226
C36	–0.035	0.001	–0.047
Eigenvalue	12.907	5.5089	3.042
Variance (%)	24.9	21.4	16.3
Source	Coal and Biomass/Vehicle mix	Vehicle	Biogenic

**Table 9 ijerph-17-01397-t009:** Seasonal and Annual average toxicity evaluation of PAHs in 2002 and 2019 (unit: ng m^−3^).

**2018**					
	Winter	Spring	Summer	Fall	Annual
BaP-TEQ	1.86 ± 2.0	0.17 ± 0.14	0.08 ± 0.09	0.28 ± 0.17	0.59 ± 1.20
BaP-MEQ	2.02 ± 1.75	0.24 ± 0.19	0.14 ± 0.13	0.43 ± 0.27	0.70 ± 1.16
China Standard ^1^	5/25 ^3^	0/28	0/23	0/32	-
WHO Standard ^2^	17/25	0/28	0/23	0/32	-
**2002**					
	Winter	Spring	Summer	Fall	Annual
BaP-TEQ	7.15 ± 4.8	4.76 ± 3.0	1.78 ± 1.0	4.57 ± 5.3	4.69 ± 4.5
BaP-MEQ	6.73 ± 4.3	3.96 ± 2.5	1.37 ± 0.8	3.28 ± 4.2	3.97 ± 4.0
China Standard ^1^	20/21	7/10	2/17	10/20	-
WHO Standard ^2^	21/21	10/10	14/17	11/20	-

^1^ Chinese NAAQS daily BaP-TEQ standard (2.5 ng m^−3^). ^2^ WHO BaP-TEQ standard (1 ng m^−3^). ^3^ Days that exceeded the standard/Number of Sampling days.

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
