# Peer review of "Recent Occurrence of PAHs and n-Alkanes in PM2.5 in Seoul, Korea and Characteristics of Their Sources and Toxicity"

_ijerph, 2020, doi:10.3390/ijerph17041397_

Round 1

Reviewer 1 Report

Given the lack of data for IPA's South Korea, the study is certainly useful.

However in the introduction (line 51) it is stated that the contribution of biomass in South Korea is substantial, in all the work this source of PAH is never taken into consideration. This affects the whole treatment and also the study of PCA. In other countries, it has been shown that in winter almost 90% of PAHs are produced by the combustion of biomass, This study does not take this source into consideration. Since the first variable justifies only 25% of the variability of the data, the PCA is rather weak.

It is therefore necessary to review the entire study taking into consideration the source of biomass burning.

In the application of the PCA it is very important to know the entity of the sources through an assessment of the emissions inventory.

The change in the concentrations of PAHs is not accompanied by an assessment of the change in sources from 2002 to 2018. It is possible, for example, that the use of coal or other fuels has decreased in the face of cleaner fuels

The sources of PAHs and alkanes  have a strong seasonality it would be necessary to apply the PCA in a different way for the season, having more data.

Meteorological variables such as temperature and radiation, which have a strong impact on PAHs, were not taken into account

(line 132) The reduction in PAHs from 2002 to 2018 is justified by the greater efficiency of combustion. The measurement of elemental carbon concentration (EC) is reported as evidence, however the reported values do not seem statistically different taking into account the standard deviation. It would also be preferable to report the value of organic carbon too. The ratio between elemental and organic carbon can give some information on the sources.

line 164 replace ratio between with per cent of

line 113 insert the type of  GC column

Author Response

Response of Reviewer 1 comments

Comment #1

However in the introduction (line 51) it is stated that the contribution of biomass in South Korea is substantial, in all the work this source of PAH is never taken into consideration. This affects the whole treatment and also the study of PCA. In other countries, it has been shown that in winter almost 90% of PAHs are produced by the combustion of biomass, This study does not take this source into consideration. Since the first variable justifies only 25% of the variability of the data, the PCA is rather weak. It is therefore necessary to review the entire study taking into consideration the source of biomass burning.In the application of the PCA it is very important to know the entity of the sources through an assessment of the emissions inventory.

Response

We agree reviewer’s suggestion that biomass is one of major sources of PAHs in the atmosphere. Thus, we added the diagnostic ratios of PAHs for biomass combustion in Table 6, however, separation of coal and biomass combustion is limited using DR of specific PAH compounds due to overlapping DR of biomass burning and coal combustion. So, in this study, instead of coal combustion, it is more proper to consider as solid fuel combustion which include the influence of both biomass and coal combustion and the limitation for separation of biomass and coal combustion was explained in line 189-193 in the revised manuscript.

“separate coal and biomass combustion and vehicular emissions. (Table 6)As seen in Table 6, separation of coal and biomass combustion is limited using DR of specific PAH compounds due to overlapping DR of biomass burning and coal combustion. So, in this study, the emission sources of PAHs can be only separated into solid fuel combustion, which includes both of coal and biomass combustion with vehicular exhaust emission.”

Table 6. Diagnostics Ratio (DR) for anlaysis.

IcdP/(IcdP+BghiP)

Fl/(Fl+Py)

BaP/BghiP

BaA/(BaA+Chr)

BaP/(BaP+BeP)

Coal combustion

0.56[35]

>0.5[37, 38]

0.9-6.6[39, 40]

>0.35[38]

Vehicular

Emission

0.18-

0.40[35, 36]

0.4-

0.5[36-38]

0.3-0.44[40]

0.2-0.35

Wood Combustion

0.62[35]

>0.5[37]

Fresh

Particle

~0.5[40]

Aging

Particle

<0.5[40]

Also, in PCA results, in order to separate the factor relating to the biomass burning from the factor for coal combustion, we think that more useful markers for biomass burning are needed such as levoglucosan. If this compound is taken into consideration, we might have a more separated factors between coal combustion and biomass burning. However, in this study, since we did not take levoglucosan into consideration, the separation of biomass burning and coal combustion was not observed from three explained factors shown in Table 8. It is challenging to estimate the impact of biomass burning on PAHs concentration in PM2.5 at Seoul apart from coal combustion in this study. So, we changed all wording for coal combustion to “solid fuel combustion” since biomass falls into solid fuel category in the revised manuscript.

Comment #2

The change in the concentrations of PAHs is not accompanied by an assessment of the change in sources from 2002 to 2018. It is possible, for example, that the use of coal or other fuels has decreased in the face of cleaner fuels

Response

We agree reviewer’s suggestion. Thus, we include increase of clean energy usage in Korea as one of possible reason to reduce PAHs concentration in PM2.5 in line 165-167 in the revised manuscript.

“Also, increase of clean fuel usage instead of coal fuel according to the policy of air quality management in Korea is one of possible reason to reduce PAHs concentration at Seoul, Korea.”

Comment #3

The sources of PAHs and alkanes have a strong seasonality it would be necessary to apply the PCA in a different way for the season, having more data.

Response

We agree the reviewer comments. Thus, we tried to apply PCA to the each season, however, PCA application was failed due to the limitation of the number of the samples for the season. For the PCA results of the winter samples failed at converging when rotated, therefore the rotated result could not be obtained. Also, PCA could not separate the summer samples. The error message reported like that “there might be less than two factors or only a single variable exists, therefore the program could not proceed to further calculations”. For the spring and fall samples, the PCA results were obtained. Spring and Fall samples set loaded 4 factors. In the PCA result of the spring samples, Factor 1(24%) grouped LMW-PAHs and lower chained n-alkanes(solid fuel combustion). Factor 2(21%) grouped longer chained n-alkanes(biogenic emission). Factor 3(19%) grouped HMW-PAHs only(vehicle emission) and Factor 4(13%) shorter chained n-alkanes. The PCA results for the fall samples also separated 4 factors. Factor 1(35%) grouped HMW-PAHs with most of longer chained n-alkanes(C26~C36). This may indicate that biogenic emission and anthropogenic emission(vehicle) influences this season together. Factor 2(27%) had most of the input PAHs and shorter chained n-alkanes, indicating a mixture of several anthropogenic sources. Factor 3(9%) highlighted HMW-PAHs (ICdP, DahA and Cor) which are markers for vehicle emission. From Factor 4(7%), sources could not be identified.

From attempt to apply PCA for the seasonal samples, we concluded that factor separation for each season was not adequate to account for the physicochemical properties of each of PAH and n-alkanes compounds. Thus, we decided to apply PCA analysis for the entire samples of PAHs and n-alkanes concentration.

Comment #4

Meteorological variables such as temperature and radiation, which have a strong impact on PAHs, were not taken into account

Response

We agree the reviewer comments. Thus, we added the meteorological information (Temperature, radiation and RH) in Table 1 and explained the seasonal difference of the meteorological factors in line 98-102 in the revise manuscript.

Detailed information on the number of samples and meteorological parames during the sampling period are given in Table 1. Average radiation, temperature and relative humidity (RH) was at maximum in summer. Radiation for winter and fall was similar, however, average temperature for winter (-4.7±6.1℃) was below the freezing point. RH increases from winter to summer”

Comment #5

Line 132: The reduction in PAHs from 2002 to 2018 is justified by the greater efficiency of combustion. The measurement of elemental carbon concentration (EC) is reported as evidence, however the reported values do not seem statistically different taking into account the standard deviation. It would also be preferable to report the value of organic carbon too. The ratio between elemental and organic carbon can give some information on the sources

Response

Thank you for the helpful comment. As reviewer suggested, we added the average OC concentration with the average OC/EC ratio for each year in Table 4 and explained the possibility of decrease of primary emission in line 153-163 in the revised manuscript.

“Organic Carbon (OC) and EC measurement have been carried out since 2013 at the same sampling site of this study, operated by the Korean National Institute of Environmental Research (NIER) as one of the intensive measurement networks of the NIER. EC concentration has shown a decreasing tendency since 2013, while, OC concentration has not been changed. Compared to the EC concentration in 2013, the EC concentration in 2018 is about half, suggesting higher combustion efficiency of fossil fuel emission sources. In addition, the OC/EC ratios which is one of indicator of secondary formation of OC in PM2.5 has been doubled from 2013 to 2018. It can be explained by an increase in secondary formation or a decreased of primary emission. The consistency of OC concentration with the increasing OC/EC ratios is primarily due to the effect of the decrease of EC concentration. Therefore, the tendency of EC concentration with OC/EC ratio can be one of evidence for the decrease of organic PM in combustion sources.”

Comment #6

Line 113: insert the type of GC column

Response

We added the GC column information with details on GC-MS operation method in line 118-123 of the revised manuscript

“with DB-5MS ultra inert column (30 m x 0.25 mm, 0.25 µm thickness, Agilent® J&W™) and quantified with a 5977A mass spectrometer (MS) in synchronous selected ion monitoring (SIM)/scan mode. Helium of 99.999% purity was used as a carrier gas at a flow rate of 1mL min-1 in a GC-MS. A 1μL sample was injected in splitless mode at 240 °C. Mass spectrometry was operated in electron impact (EI) mode at 70 eV at a source temperature of 230 °C.”

 Comment #7

Line 164: replace the ratio between with per cent of

Response

Thank you for the comment. We replaced the expression “Ratio between” as “Percent of” in Figure 2.

Reviewer 2 Report

Lee and coworkers report on the recent occurrence of PAHs and n-Alkanes in ambient aerosol in Seoul, Korea and the characteristics of their sources. The study reports the recent PAHs concentration in the particulate matter with an aerodynamic diameter of less than or equal to a nominal 2.5 μm (PM2.5) with the index of Cmax, CPI and WNA(%) in n-Alkanes in 2018. The seasonal differences in the total concentration of PAHs and n-Alkanes were clear where winter showed visibly higher concentration than that of warmer seasons. Coal and vehicular exhaust were identified as the major emission sources of PAHs in PM2.5 in the winter, while coal combustion was not significant in summer. This study provides important insight into the composition of pollutants including temporal data, both of which may aid in the formulation of new environmental & public health policies for achieving clean air in Seoul, Korea.

The introduction is well written and contains references to key background information. The authors do a good job of explaining the carbon preference index (CPI), obtained from the ratio of the sum of odd numbered n-alkanes to the sum of even numbered n-alkanes, as well as the expected CPI values if alkane emission is arising purely from anthropogenic or biogenic sources. The analytical procedure of how the samples were collected, analyzed, and measured using GC/MS is well described for the most part (see minor edit request below). The data were well presented in the both tabular and graphical forms as appropriate. Principle Component Analysis (PCA) was used to group measured species with common variances, i.e., principle components and the PCA was used to identify sources of air pollutants. Species showing higher values for each factor were considered to share an origin. Back trajectory analysis was utilized to determine the sources of these pollutants (northwest of the Korean peninsula). Overall data shows decrease in toxicity and mutagenicity of the overall blend of pollutants in the aerosol mixtures between the 2002 and 2008 samples, although the pollutant levels still exceed the WHO standards during winter months – a season mostly affected by burning coal as well as vehicular emissions. This study suggests that control of the PM2.5 toxicity in Seoul needs to be focused on the regulation of PAHs emission sources in winter season.

Overall the article is well written, contains useful data and analysis on air pollutants in Korea. The manuscript is of interest to the readership of the International Journal of Environmental Research and Public Health and warrants publication after minor revisions described below.

Line 114-115: The authors need to specify the column using on the GC/MS instrument as well as the flow rate and identity of the mobile phase used for separating/quantifying the various components of the pollutant sample.

Line 204: it is not clear that the acronym “SPSS” in this sentence is referring to a software package. Consider rephrasing this sentence and/or clarifying what SPSS is.

Line 289: spelling mistake, change “toxitity” to “toxicity”

Author Response

Response of Reviewer 2 comments

Comment #1

Line 114-115: The authors need to specify the column using on the GC/MS instrument as well as the flow rate and identify of the mobile phase used for separating/quantifying the various components of the pollutant sample.

Response

We added the GC column information with details on GC-MS operation method in line 118-123 of the revised manuscript

“with DB-5MS ultra inert column (30 m x 0.25 mm, 0.25 µm thickness, Agilent® J&W™) and quantified with a 5977A mass spectrometer (MS) in synchronous selected ion monitoring (SIM)/scan mode. Helium of 99.999% purity was used as a carrier gas at a flow rate of 1mL min-1 in a GC-MS. A 1μL sample was injected in splitless mode at 240 °C. Mass spectrometry was operated in electron impact (EI) mode at 70 eV at a source temperature of 230 °C.”

Comment #2

Line 204: It is not clear that the acronym “SPSS” in this sentence is referring to a software package. Consider rephrasing this sentence and/or clarifying what SPSS is.

Response

Thank you for the comment. We clarified SPSS as the SPSS 18.0 statistical software in line 231

Comment #3

Line 289: spelling mistake, change “toxitity” to “toxicity”

Response

Sorry about it. The typo was corrected in the revised manuscript.

Reviewer 3 Report

1.Please provide Scores and Loadings Plots for PCA analysis.

2. Include experimental details about the PCA computation method and environment used.

3. Please provide a Scree plot on how the number of components were chosen for the PCA analysis.

4. Please provide details on what pre-processing methods were used and the rationale behind that.

5. Also, include a short summary of the PCA method in the Introduction to make it more appealing to a wider audience.

Author Response

Response of Reviewer 3 comments

Comment #1
Please provide Scores and Loading Plots for PCA analysis
Response
As reviewer suggested, we added Score plot and Loading plots of PCA results in Figure 7 in the revised manuscript and explained the distribution of each compounds in the loading plot with the distribution of seasonal samples in score plot in line 246-256 of the revised manuscript.

“Figure 7 shows both the loading and score plot of PCA analysis for three factors. Winter and other seasons’(spring, summer, fall) samples were well separated (Figure 7 (a), (b)) which indicates different emission sources. The lower right part of the score plot is dominated by winter samples, which is characterized by LMW-PAHs, HMW-PAHs and lower chained n-alkanes in the loading plot (Figure 7 (b)). Therefore a mixture of solid fuel combustion and vehicle emission is a major influence on winter atmosphere. Fall samples mostly overlapped with spring and summer samples in the lower left corner of the score plot, which implies the samples of these seasons has the similar emission source. The overlapped area of the three seasons all included HMW-PAHs (DahA, Cor) and longer chained (C>25) in the loading plot. Thus we can expected that spring, summer and fall were mainly influenced by biogenic emissions and vehicular emission.”

(a)

(b)

(c)

Figure 7. Loading plots and Score plots of (a) Factor 1 and Factor 2 (b)Factor 1 and Factor 3 (c)Factor 2 and Factor 3.

Comment #2
Include experimental details about the PCA computation method and environment used. Please provide details on what pre-processing methods were used and the rationale behind that.
Response
We added details on how PCA is applied from the data of individual PAH compounds and n-alkanes concentrations in PM2.5 samples to clarify the process of PCA analysis in line 227-240 in the revised manuscript. In addition, Table 8 was modified to show the PCA results clearly.

“PCA was performed to group measured species with common variances, i.e., principal components. Normally, the first principal component would represent the highest variance, followed by the second and third, and so on. Species showing higher values for each factor were considered to share an origin. For PCA, the concentration of 14 PAH and 16 n-alkane compounds were applied as variables into the SPSS 18.0 statistical software for 108 total PM2.5 samples. Varimax rotation was used to get as many positive loadings as possible to achieve a more meaningful and interpretable solution for air pollutants data suggested from previous studies [reference]. N-alkanes were accounted into two groups: low (∑C20-C25) and high (∑C26-C36) molecular weight (MW) to characterize fossil fuel combustion and biogenic emission, respectively. For 14 PAHs, each ring-group of PAHs were used as indicator to distinguish the combustion sources of solid fuel (coal and biomass) and liquid fuel (vehicular emission). Combustion of solid fuels such as coal and biomass mainly emit PAHs that have three and four benzene rings, and liquid fuel combustion such as vehicular emissions is a source for PAHs with five or more benzene ring [18-21]. Sensitivity analysis by number of variable was conducted in PCA and those results were found to be relatively consistent.”
Comment #3
Please provide a Scree plot on how the number of components were chosen for the PCA analysis.
Response
Thank you for valuable suggestion. We added scree plot in Figure 6 in the revised manuscript and checked the point of the rapid change of steep slope after flat out between Eigen value and components. In PCA, generally Eigen value larger than 1 is selected for valuable components, however, in scree plot, if the rapid change of slope were appeared at some component, we can select up to this component to describe the data. We added details on the selection of the number of factors with scree plot in line 241-245 in the revised manuscript.
“Five factors were extracted as the results of eigenvalue larger than 1. Generally Eigen value larger than 1 is chosen for valuable components, however, the rapid change of slope were appeared at component 4 from scree plot shown in Figure 6. Thus, three components were valuable to identify sources of PAHs and n-Alkanes in PM2.5. Indeed, variance for Factors 4 and 5 were 7% and 6%, respectively, which was insignificant compared to Factor 1 (25%), Factor 2 (21%), and Factor 3 (16%).”

Figure 6. Scree plot of the Principle Component Analysis

Comment #4
Include a short summary of the PCA method in the Introduction to make it more appealing to a wider audience.
Response
We included introduction of PCAs relating to the advantage and application to the air pollutants in the introduction section (line 80-84) of the revised manuscript
“Principle Component Analysis (PCA) is an effective statistical tool to identify independent factor of air pollutants by grouping chemical species which share similarities of variances of the species to give physicochemical significance to these groups. PCA has been widely used to the sources identification of air pollutants in PM2.5, [31,32].”
